# Assessment of the Antioxidant and Hypolipidemic Properties of *Salicornia europaea* for the Prevention of TAFLD in Rats

**DOI:** 10.3390/antiox13050596

**Published:** 2024-05-12

**Authors:** Aymen Souid, Lucia Giambastiani, Antonella Castagna, Marco Santin, Fabio Vivarelli, Donatella Canistro, Camilla Morosini, Moreno Paolini, Paola Franchi, Marco Lucarini, Andrea Raffaelli, Lucia Giorgetti, Annamaria Ranieri, Vincenzo Longo, Luisa Pozzo, Andrea Vornoli

**Affiliations:** 1Department of Agricultural, Food and Agro-Environmental Sciences, University of Pisa, Via del Borghetto 80, 56124 Pisa, Italy; souid.ayman@gmail.com (A.S.); antonella.castagna@unipi.it (A.C.); marco.santin@unipi.it (M.S.); anna.maria.ranieri@unipi.it (A.R.); 2Institute of Agricultural Biology and Biotechnology—National Research Council (IBBA-CNR), Via Moruzzi 1, 56124 Pisa, Italy; lucia.giambastiani@gmail.com (L.G.); andrea1.raffaelli@santannapisa.it (A.R.); lucia.giorgetti@cnr.it (L.G.); vincenzo.longo@cnr.it (V.L.); andrea.vornoli@cnr.it (A.V.); 3Department of Pharmacy and Biotechnology, Alma Mater Studiorum—University of Bologna, Via Irnerio 48, 40126 Bologna, Italy; fabio.vivarelli3@unibo.it (F.V.); donatella.canistro@unibo.it (D.C.); camilla.morosini2@unibo.it (C.M.); moreno.paolini@unibo.it (M.P.); 4Department of Chemistry “G. Ciamician”, Alma Mater Studiorum—University of Bologna, Via S. Giacomo 11, 40126 Bologna, Italy; paola.franchi@unibo.it (P.F.); marco.lucarini@unibo.it (M.L.); 5Crop Science Research Center, Scuola Superiore Sant’Anna, Piazza Martiri della Libertà 33, 56127 Pisa, Italy

**Keywords:** *Salicornia europaea*, antioxidants, steatosis, cytochrome P450, polyphenols

## Abstract

Halophyte species represent valuable reservoirs of natural antioxidants, and, among these, *Salicornia europaea* stands out as a promising edible plant. In this study, young and old *S. europaea* leaves were compared for the content of bioactive compounds and antioxidant activity to assess changes in different growth phases; then, the potential protective effects against low-dose CCl_4_-induced toxicant-associated fatty liver disease (TAFLD) were investigated by administering an aqueous suspension of young leaves to rats daily for two weeks. Quantification of total and individual phenolic compounds and in vitro antioxidant activity assays (DPPH, FRAP, and ORAC) showed the highest values in young leaves compared to mature ones. Salicornia treatment mitigated CCl_4_-induced hepatic oxidative stress, reducing lipid peroxidation and protein carbonyl levels, and preserving the decrease in glutathione levels. Electronic paramagnetic resonance (EPR) spectroscopy confirmed these results in the liver and evidenced free radicals increase prevention in the brain. Salicornia treatment also attenuated enzymatic disruptions in the liver’s drug metabolizing system and Nrf2-dependent antioxidant enzymes. Furthermore, histopathological examination revealed reduced hepatic lipid accumulation and inflammation. Overall, this study highlights Salicornia’s potential as a source of bioactive compounds with effective hepatoprotective properties capable to prevent TAFLD.

## 1. Introduction

Given the escalating global climate change and the difficult conditions prevailing worldwide, the cultivation of traditional crops is facing numerous constraints [1,2]. These challenges encompass the scarcity of high-quality water, rising temperatures, salinization, heavy metal contamination, and the deterioration of soil properties, as occurs in arid and semi-arid regions of the Mediterranean basin [3,4]. The excessive use of chemicals such as fertilizers, pesticides, and herbicides has significantly boosted crop yields and global food production as well as posing potential risks to human health, as residues of these chemicals can persist on crops. Additionally, the leaching of chemicals into groundwater can contaminate drinking water sources. Beyond immediate health concerns, the environmental impact of chemical use can indirectly affect human health through altered ecosystems and food quality [5]. Unfortunately, these dangerous environmental pollutants and toxic chemicals such as aflatoxins, carbon tetrachloride (CCl4), perfluorinated alkylated substances, and drugs cause health problems, and especially liver damage [5,6]. The liver serves as the primary organ responsible for metabolizing and detoxifying xenobiotics, making it susceptible to various harmful conditions that can impair its crucial functions [7]. This predisposition can lead to life-threatening disorders, including steatosis, hepatitis, cirrhosis, hepatic failure, and, in particular, hepatocellular carcinoma [8]. The Toxicant-Associated Fatty Liver Disease (TAFLD) primarily results from exposure to harmful substances or chemicals [6,8,9]. This condition occurs when the liver encounters environmental toxins, such as industrial chemicals, pesticides, heavy metals, and air pollution, which can cause damage and led to the development of fatty liver disease [8]. Some occupations, particularly those involving contact with toxic materials, increase the risk of exposure, making individuals in sectors such as manufacturing, mining, and agriculture more susceptible to TAFLD. Toxicants can enter the body through various routes, including ingestion and inhalation, such as consuming contaminated food or water or inhaling polluted air. Additionally, certain medications and food toxins can contribute to liver damage and the onset of TAFLD [9]. CCl_4_ was extensively utilized as a solvent, vermicide, refrigerant, and in fire extinguishers. However, its applications were curtailed following the reporting of several hundred cases of toxicity in the first-half of the 20th century [10]. It persisted as a feedstock for chlorofluorocarbon (CFC) synthesis, but experienced a reduction in U.S. production to 130 million pounds per year after the implementation of the Clean Air Act and the Montreal Protocol, which banned CFCs [11]. In any case, CCl_4_ is stable and environmentally persistent, resulting a pervasive ambient air pollutant and can contaminate aquifers. Estimated daily intake by the general U.S. population from air and water sources ranges from 12 to 511 μg/day and 0.2 to 60 µg/day, respectively [8]. As hepatotoxic substance, CCl_4_ is capable of inducing acute liver damage, and was the first chemical used to create a toxic fatty liver model. Through the generation of activated oxygen-free radicals, CCl_4_ accelerates the progression of fatty degeneration and fibrosis, leading to the disruption of hepatocellular structure and function. CCl_4_ metabolites prompt Kupffer cells to release proinflammatory cytokines, further exacerbating liver damage [12]. In a previous study performed in rats, CCl_4_ was administered intragastrically, resulting in significant increases in liver fat droplets, liver damage (alanine transaminase (ALT), and aspartate transaminase (AST)), and DNA damage (8-Hydroxyguanosine (8-OHdG)), with a concomitant decrease in the antioxidant enzymes superoxide dismutase (SOD), glutathione (GSH), glutathione peroxidase (GPX), and catalase (CAT) in the liver due to their rapid depletion after fighting oxidative stress [13]. Low-dose intraperitoneal injections of CCl4 in C57BL/6 mice fed a CDAA diet elicited marked non-alcoholic steatohepatitis (NASH) features compared to the CDAA diet alone and gave rise to hepatocellular carcinoma (HCC) [14]. In general, these studies revealed to be strictly dependent on exposure time and dosage, since CCl4 can induce fatty liver very rapidly. It is essential to investigate the repercussions of low-level exposure to toxic substances as, even at minimal concentrations encountered in everyday life, it has the potential to induce steatosis and liver injury [6]. Globally, liver disease has emerged as a rapidly escalating health crisis, characterizing by rising mortality rates. Adding to the distress of affected patients, existing treatment modalities, which include drug therapy and liver transplantation, show limited efficacy and are marred by inherent risks [8,9]. These challenges have spurred the exploration of safer and more effective therapeutic alternatives, with particular attention to natural sources. In this context, halophyte plants and their bioactive secondary metabolites have garnered considerable interest due to their immense potential in the management and improvement of various liver disease [15,16,17]. Halophyte plants have evolved adaptive mechanisms to cope with the production of highly reactive oxygen species (ROS), which are toxic to cellular structures [18]. These adaptations include the synthesis of potent bioactive compounds known for their robust antioxidant properties, such as phenolic compounds, antioxidant enzymes, and vitamins [18,19]. These natural antioxidants exhibit impressive biological activities, including the ability to scavenge free radicals, bind to metals, and induce beneficial enzymes, contributing to their therapeutic potential. As a result, some halophytic species have found application in various traditional dishes, recipes, and drinks. Furthermore, with recent advances in food science and technology, there is a growing need to assess the safety profiles of these plants through metabolomic and toxicological studies to ensure the production of safe and healthy foods [3,19,20,21]. *Salicornia europaea* L., commonly known as sea beans, is a halophyte plant that thrives along the coasts of America, Europe, South Africa, and the Mediterranean basin. Research conducted on *S. europaea* extracts from Mediterranean regions has revealed a substantial presence of phenolic and flavonoid compounds, which have been proposed as contributors to its notable antioxidant capabilities [22,23]. Additionally, *S. europaea* extracts have been traditionally employed in the Eastern pharmacopeia for the treatment of various ailments, suggesting its potential utility in several industrial and biotechnological sectors [22,23]. Despite previous recognition of the abundance of phenolic compounds and antioxidant properties in *S. europaea* extracts, to the best of our knowledge, there is a lack of information on the antioxidant capacity in the differential growth phases and its beneficial effects in vivo. Accordingly, a comparison of total and individual phenolic compounds and pigments of hydro-methanolic extracts of young and old *S. europaea* leaves was performed in the present study. Then, the potential protective effects of an aqueous suspension of young *S. europaea* leaves against low dose CCl4-induced hepatotoxicity in rats were investigated, particularly for its ability to ameliorate the TAFLD pathology.

## 2. Materials and Methods

### 2.1. Plant Material and Methanolic Extraction

*S. europaea* young and mature (old) leaves were collected from the research center greenhouse at the University of Pisa in Rottaia. The plant collection samples were gathered by considering the plant’s visual characteristics and growth stage. Young Salicornia shoots (leaves) are usually tender, slender, and green, boasting an elegant appearance that makes them a sought-after choice for culinary use. In contrast, adult shoots are more mature, thicker, and may develop a woody texture, often acquiring reddish or brownish hues with age. The leaves underwent a thorough cleansing process using deionized water, followed by immediate soaking. Subsequently, they were preserved at −80 °C and then subjected to freeze-drying (Scanvac CoolSafe LaboGene, Lillerød, Denmark). The resulting dry material was finely powdered and stored at −20 °C until it was ready for analysis. A gram of powdered young or mature (old) Salicornia leaves was subjected to maceration in 10 mL of 80% methanol for an overnight extraction, with continuous stirring at room temperature [24]. Following this, the mixtures were centrifuged at 3500× *g* for 10 min at 4 °C using a Jouan CR3i centrifuge (Newport Pagnell, Milton Keynes, UK). The resulting supernatants were then collected and stored at 4 °C in darkness until required for use. This extraction process was repeated twice on the residual pellets. Both for young and mature leaves, the resulting three filtered supernatants were combined.

The extracts obtained from young and mature Salicornia leaves were both analyzed and compared for the bioactive molecules and the individual phenolic compounds content and for the antioxidant activities.

### 2.2. Antioxidant Profiling of Young and Mature S. europaea Leaves

#### 2.2.1. Bioactive Molecules Content

The Folin–Ciocalteu colorimetric method of Singleton et al. [25] was employed to assess the overall phenolic content, reported as mg of gallic acid equivalents per gram of dry weight (mg GAE/g DW). The total flavonoid content was determined using the aluminum chloride colorimetric method of Kim et al. [26] and expressed as mg catechin equivalent per gram of dry weight (mg CE/g DW). The quantification of total flavonols, measured in terms of mg quercetin equivalent per gram of dry weight (mg QE/g DW), was conducted according to the procedure outlined by Romani et al. [27]. Total monometric anthocyanins were quantified using the differential pH spectrophotometric method described by Lee et al. [28] and expressed as mg cyanidin-3-glucoside (C3GE)/g DW. The quantification of chlorophyll A and B and carotenoids was performed utilizing the method outlined by Lichtenthaler et al. [29]. The concentrations of pigments were calculated according to the following formulas and were expressed as µg/g of FW.:Chlorophyll a = (12.7 × A663) − (2.69 × A645)
Chlorophyll b = (22.9 × A645) − (4.68 × A663)
Total chlorophylls = (20.2 × A645) + (8.02 × A663)
Carotenoids = (5 × A460) − ((Chl a × 3.19) + (Chl b × 130.3))

#### 2.2.2. Phenolic Compounds Profiling by UHPLC-ESI-MS/MS Analysis

Well-known phenolic compounds were selected for a thorough quantitative analysis of the extracts using a Sciex 5500 QTrap+ mass spectrometer (AB Sciex LLC, Framingham, MA, USA) via Ultra-High-Performance Liquid Chromatography with Electrospray Ionization Tandem Mass Spectrometry (UHPLC-ESI-MS/MS). The mass spectrometer was equipped with a Turbo V ion-spray source and linked to an ExionLC AC System, custom-made by Shimadzu (Shimadzu Corporation, Kyoto, Japan), which consists of two ExionLC AC pumps, an autosampler, a controller, a degasser, and a tray. MS/MS experiments were conducted in the electrospray negative ion mode with nitrogen serving as the collision gas. The operational source parameters included source type as turbospray, nebulizer gas (GS1) at 70, turbo gas (GS2) at 50, curtain gas (CUR) at 10, temperature (TEM) at 500 °C, Ionspray Voltage (IS) at −4500 V, and entrance potential (EP) at 10 V. Compound parameters such as declustering potential (DP), collision energy (CE), and collision cell exit potential (CXP) were fine-tuned for the specific Selected Reaction Monitoring (SRM) transition of each component. The analyses were carried out in triplicate, and the results are presented as micrograms per 100 g of dry weight.

#### 2.2.3. In Vitro Assays to Measure Antioxidant Activity

The in vitro antioxidant potential of the two methanolic extracts of *S. europaea* leaves was assessed employing a composite approach involving fluorimetric and spectrophotometric methodologies. The evaluation of the 2,2-diphenyl-1-picrylhydrazyl (DPPH) radical scavenging activity was conducted using the method described by Sokmen et al. [30]. The antiradical activity (ARA) was computed as the percentage of DPPH• inhibition using this equation: ARA % = [1 − (AS/AC)] × 100, where AS represents the absorbance of the sample and AC denotes the absorbance of the DPPH solution.

The determination of the Oxygen Radical Absorbance Capacity (ORAC) for the *S. europaea* leaf extract followed the procedure outlined by Bacchiocca et al. [31]. Thoroughly, AAPH was used as a peroxyl radical generator, with fluorescein as a probe, and Trolox as the standard antioxidant. The findings were expressed in ORAC units (µmol of Trolox equivalents (TE) per g of dry weight (dw)).

To determine the antioxidant capacity of *S. europaea* leaves extract, we utilized the Ferric Reducing Antioxidant Power (FRAP) assay detailed by Benzie et al. [32], with some modifications. A solution comprising acetate buffer, TPTZ and FeCl_3_·6H_2_O was combined with the extract. After incubating for 6 min at room temperature, absorbance was measured at 593 nm using a spectrophotometer (Perkin-Elmer Lambda 365, Shelton, CT, USA). Results were presented herein as the EC50 (effective concentration) of the samples, expressed in mg/mL.

### 2.3. In Vivo Experiment: Hepatoprotective Assay

#### 2.3.1. Animal Procedure

The in vivo experiment was performed using forty male Wistar rats of about 200 g body weight (bw). The animals were divided into four groups, housed in cages subjected to a 12 h light and dark cycle at room temperature with a relative humidity of 55%, and provided with unrestricted access to drinking water and food. A standard feed was administered in pellet, containing 19.0% of proteins, 6.0% of fibers, 7.0% of minerals and vitamins moisture, 64.0% of carbohydrates, and 4.0% of fats. Based on our analyses, young *S. europaea* leaves turned out to be richer in bioactive molecules and with greater antioxidant power than mature leaves; therefore, the former were selected for treating animals in the in vivo experiment. The four groups, of ten animals each, were divided as follows: (1) control rats (CTR), (2) rats supplemented daily by gavage during 14 days with a water suspension of *S. europeae* leaves at the dose of 300 mg/kg bw (SAL), (3) rats injected intra-peritoneally (i.p.) with a single dose of 0.8 mL/kg bw CCl_4_ dissolved in corn oil and sacrificed after 24 h (CCl_4_), and (4) rats supplemented daily during 14 days with a water suspension of *S. europaea* leaves at the dose of 300 mg/kg bw, followed on the fourteenth day by a single i.p. injection of 0.8 mL/kg bw CCl_4_ (50% in corn oil) (CCl_4_ + SAL). The weight of each animal was noted at the beginning of the experiment, after one week following initiation, and finally at the experiment’s conclusion, coinciding with the sacrifice of the animals. Before the sacrifices, blood samples were taken from each animal of the three experimental groups by cardiac puncture under general anesthesia, and then centrifuged at 2200 rcf for 15 min to obtain plasma samples for later laboratory analysis. The final sacrifice of experimental animals was performed by heart removal after blood collection. Liver tissues were weighted and stored at −80 °C for extraction and quantification of hepatic lipids, microsomes preparation, evaluation of biochemical oxidative stress markers or preserved in a 70% mixture of ethyl and isopropyl alcohol (respectively, approximately 60% and 40%) and 30% distilled water solution at 4 °C for histopathological analysis [6]. Samples of both brain and liver tissues were weighted and stored at −80 °C for analysis of reactive oxygen, nitrogen, and carbon species (ROS, RNS, and RCS) using electronic paramagnetic resonance (EPR) spectroscopy. All animal procedures of the present experiment (Autorization N. 32/2023-PR (protocol N. 65E5B.68)) were performed with the approval of the Local Ethical Committee according to the Italian law regulating the use and humane treatment of animals for scientific purposes (decreto legislativo 26/2014), and the European Union Directive 2010/63/EU for animal experiments.

#### 2.3.2. Analysis of Plasma Biochemical Parameters

The concentrations of total cholesterol (TC), high-density lipoprotein cholesterol (HDL-C), low-density lipoprotein cholesterol (LDL-C), triglycerides (TG), urea, creatinine, and the enzymatic activities of ALT and AST were measured in accordance with the manufacturer’s instructions, utilizing a commercial assay made in specialized laboratory (PAIMBiolabor, Livorno, Italy). Cardiovascular risk indices were calculated using the following formulas: cardiovascular risk index 1 = TC/HDL-C, and cardiovascular risk index 2 = LDL-C/HDL-C, as described by [33]. The antiatherogenic index (AAI) was calculated using the formula: AAI = HDL-C × 100/(TC − HDL-C) [34].

#### 2.3.3. Analysis of Reactive Radical Species (RRS) Centered on Oxygen, Nitrogen, and Carbon in the Liver and Brain Biopsies Using Electronic Paramagnetic Resonance (EPR) Spectroscopy

Biopsies weighing about 150 mg were treated with 0.5 mL of standard physiological solution containing the hydroxylamine “spin probe” (bis(1-hydroxy-2,2,6,6-tetramethyl-4-piperidinyl) decandioate dihydrochloride), synthetized as previously reported by Vivarelli et al. [35], using (1 mM) and deferoxamine (1 mM) as metal chelating agent. The tubes were incubated at 37 °C for 5 min, and then the nitroxide spectra generated by the reaction of the probe with the radicals produced in the tissues were recorded using the following parameters: modulation amplitude = 1.0 G; conversion time = 163.84 ms; modulation frequency 100 kHz; and microwave power = 6.4 mW. The intensity of the first spectral line of the nitroxide (aN = 16.90 G and g = 2.0056) was considered as a nitroxide amount in each examined adjusted for the weight of the biopsy. The calibration of the spectrometer response was performed by using a known solutions of TEMPO-coline in water and an ER 4119HS Bruker Marker Accessory as internal standard [36].

#### 2.3.4. Biomarkers of Oxidative Stress in the Liver

The malondialdehyde (MDA) concentration in liver samples was assessed following the method outlined by Seljeskog and collaborators [37], with minor adjustments as detailed in our previous publication [38]. MDA concentration was quantified in nanomoles per gram of tissue. The level of protein oxidation was determined using the carbonyl protein assay based on the protocol by Terevinto et al. [39], with slight modifications as described in Pozzo et al. [38]. Concentrations of carbonylated proteins were calculated in nanomoles per micrograms of proteins. GSH levels were measured using the method established by Browne and Armstrong [40], with slight modifications as reported in Pozzo et al. [38], and GSH concentrations were expressed in micromoles per gram of tissue.

#### 2.3.5. Assessment of Hepatic Xenobiotic Metabolizing Enzyme Activities

The microsomal and cytosolic fractions of the liver were isolated following the procedure outlined by Longo et al. [41]. Protein concentrations were assessed as per the method described by Bradford et al. [42], with bovine serum albumin (BSA) used as the reference standard. The effects of Salicornia on xenobiotic metabolizing cytochrome P450 (CYP) enzymes were investigated analyzing the ethoxycoumarin O-deethylase (ECOD) and p-nitrophenol (pNPH) specific activities. The determination of ECOD was conducted through the quantification of umbelliferone formation, following the method described by Aitio et al. [43]. The results were expressed as pmol umbelliferone/(mg protein × min). Furthermore, the pNPH activity was ascertained by monitoring the formation of p-nitrocatechol following the method of Reinke et al. [44] and results expressed as nmol/(mg protein × min). The activity of heme oxygenase-1 (HO-1) was assessed using the procedure described by Naughton et al. [45] quantifying bilirubin production through the reduction of biliverdin. Measurements were taken based on the variance in absorbance between 464 and 530 nm (with an extinction coefficient, ε, of 40 mM^−1^ cm^−1^), and the calculated HO-1 activity was expressed in picomol/(mg prot × minute). DT-diaphorase activity was determined by monitoring the reduction of dichlorophenolindophenol at 630 nm following the method described by Benson et al. [46]. The results were reported as nmol/(mg protein × min).

#### 2.3.6. Quantitation of Liver Lipids

Hepatic lipids content was determined using the gravimetric method by Folch et al. [47], slightly modified. Rat liver samples were homogenized with equal volumes of water and methanol. The resultant homogenate underwent three sequential extractions with chloroform, followed by two rinses with 1 M KCl and water. Following thorough evaporation and extended drying of the chloroform solution (until a constant weight was achieved), lipid content was weighed and expressed as mg/g tissue.

#### 2.3.7. Histopathological Analysis

After death, all experimental animals were subjected to necropsy following laboratory standard operating procedures (SOPs). Livers were collected and preserved in a 70% solution of alcohols (a mixture of ethyl and isopropyl alcohol, respectively, approximately 60% and 40%), and 30% distilled water. Trimming was performed according to SOPs. Each trimmed liver specimen was processed and embedded in paraffin blocks according to laboratory standard SOP using an inclusion control unit Bec150 (Bio-Optica, Milano, Italy). Then, 5 µm sections were cut using a microtome Supercut 2050 (Reichert-Jung, NY, USA) and routinely stained with Hematoxylin-Eosin (HE). Histopathology evaluation was systematically performed as blinded evaluation, without prior knowledge of the groups. For pathological diagnoses, the same evaluation criteria and the same classification based on INHAND (International Harmonization of Nomenclature and Diagnostic Criteria for Lesions in Rats and Mice) guidelines, were adopted by Thoolen et al. [48]. A staging score was developed to assess the degree of both severity (minimal, mild, or severe) and extent (diffuse or focal) of inflammatory, necrotizing, or other degenerative lesions such as fatty change. The scoring system described was based on the main histological features for the diagnosis of lesions attributable to TAFLD and/or toxicant associated steatohepatitis (TASH) in humans [49]. In fact, rodent models successfully simulate the histological patterns of the disease, even if they cannot reflect all the clinical and etiological aspects of the human counterpart.

### 2.4. Statistical Analysis

The results are presented as the mean (n = 3) value ± standard deviation (SD). Significant differences between means of the two types of extract from Salicornia leaves (young vs. mature) were evaluated performing Student’s *t*-test with significance at *p* ≤ 0.05. In the in vivo experiment, significant differences between means of the four rat groups were analyzed through a one-way analysis of variance (ANOVA) and a Tukey post hoc test with significance at *p* ≤ 0.05. Analyses were performed using Prism, GraphPad Software, located in San Diego, CA, USA.

## 3. Results and Discussion

### 3.1. Bioactive Compounds and Antioxidant Capacity of Salicornia

A comparative analysis was conducted on the bioactive compounds and antioxidant activity in the two freeze-dried methanolic extracts derived from young and mature leaves of *S. europaea*. Our investigation aimed to discern variations between leaves at distinct developmental stages (Table 1). Young leaves displayed significantly higher levels of bioactive compounds, such as total polyphenols, flavonoids, flavonols, and anthocyanins (8.53 mg GAE/g DW, 0.51 mg CE/g DW, 0.11 mg QE/g DW, and 0.18 mg C3GE/g DW, respectively), compared to their older counterparts (4.26 mg GAE/g DW, 0.14 mg CE/g DW, 0.05 mg QE/g DW, and 0.10 mg C3GE/g DW, respectively). In the existing literature, the assessment of phenolic compounds of Salicornia at various plant ages has not been considered so far. However, there are studies that have generically examined the content of phenolic compounds in Salicornia and other halophytic plants. In a work by Cristina Costa et al. [50] aimed to test the extraction efficacy with different ethanol concentrations in *S. europaea*, total polyphenols were between 1.3 to 9.3 mg GAE/g DW; these results are in the same range as those of our work, despite the adoption of different extraction techniques. Similar results were found in *S. europaea* extracts collected from different regions in the southeastern coast of Tunisia (Gabés and Boughrerra) [23], demonstrated that Salicornia leaves showed a high content of secondary metabolites. Comparable results were observed in several halophyte plants such as *Inula Crithmoides*, *Limonium* sp. collected from different regions of Tunisia [51] and *Crithmum maritimum* plants collected from the coast of Portugal and during summer season from the cliffs of Brittany in France [17,21,52]. Additionally, young Salicornia leaves exhibited significantly higher levels of chlorophyll a and b, and increased, although not significantly, the levels of carotenoids (465.3 µg Chl a/g FW, 233.4 µg Chl b/g FW, and 106.7 µg /g FW, respectively), compared to mature leaves (407.8 µg Chl a/g FW, 157.8 µg Chl b/g FW, and 102.6 µg/g FW, respectively), which are key pigments responsible for photosynthesis, antioxidant activities, and contribute to vibrant colors. A recent study by Castagna et al. [22] reported that *S. europaea* showed very similar pigment concentrations both when the plant was collected from natural biotope and under greenhouse conditions. Furthermore, as assessed by DPPH, FRAP, and ORAC assays, the antioxidant activity of Salicornia extracts was also significantly more pronounced in those obtained from young leaves (79.51% ARA, EC50 6.79 mg/mL, and 268.08 µmol TE/g DW, respectively), compared to that obtained from the mature ones (60.6% ARA, 3.4 EC50 mg/mL, and 164.9 µmol TE/g DW, respectively), underlining the greater potential in neutralization of free radicals. Other studies have documented similar high antioxidant properties in extracts of *S. europaea* and other halophytes species [22,23,52]. Interestingly, the values obtained were much greater than those reported in one of our recent investigations involving a leafy variety of *Brassica oleracea*, a species well known for its great antioxidant properties, in which we found a mean ORAC value of 14.9 μmol TE/g dw. [53]. All in all, these analyses suggest that the younger leaves are richer in bioactive compounds and pigments and, consequently, have a stronger antioxidant activity compared to their mature counterparts, thus resulting much more suitable for being used in the in vivo experimental study to test its potential health benefits.

### 3.2. Quantification of Polyphenols by UHPLC-ESI-MS/MS

In order to obtain a more comprehensive understanding of the phenolic composition of *S. europaea* at different growth stages, the principal constituents and their respective quantities were elucidated using UHPLC-ESI-MS/MS. Among the twenty-seven phenolic compounds identified, eight were recognized as phenolic acids, while the remaining nineteen were categorized as flavonoids, as detailed in Table 2. As expected, given the differences emerged from bioactive compounds analysis, this investigation showed intriguing variations between young and old leaves extracts. The most abundant compound identified in both the extract was unequivocally the 3-O-Caffeoylquinic acid (chlorogenic acid), which showed a significantly higher level in young leaves compared to the old ones (1984.85 vs. 865.20 µg/g DW, respectively). Other phenolic acids such as gallic acid, caffeic acid, p-coumaric acid, vanillic acid, trans-ferulic acid, rosmarinic acid, and protocatechuic acid also exhibited higher levels in young leaves compared to mature leaves (Table 2). These results are in contrast to what observed by Kim et al. [54] in another species of Salicornia; in fact, *Salicornia herbacea* L. exhibited higher total phenolic content during the maturity stage compared to the initial growth stage. Nevertheless, our results align with previous research on Tunisian ecotypes of *S. europaea*, confirming that chlorogenic acid, hydroxycinnamic acid, and rutin hydrate were the predominant phenolic compounds in leaves [23]. Among the detected flavonoid compounds, our analysis revealed that flavonols represented the most abundant in the *S. europaea* extract, with a noteworthy prevalence of 3-O-quercetin derivatives as the primary flavonoid aglycones. In particular, quercetin 3-O-glucoside was the second most represented phenolic compound in both young and old leaves, albeit in the early stage its amount was more than double compared to the latter (154.33 vs. 61.99 µg/g DW, respectively). Comparable results in phenolic compounds profiling were observed in other studies on fresh, oven-dried, and freeze-dried extracts of *Salicornia ramosissima* [55], *Salicornia patula* [56], and *Salicornia gaudichaudiana* [57]. Thanks to its phytochemical composition, Salicornia can promote health by influencing glucose and lipid metabolism and protecting against damage from ROS [58]. Our comparative analysis underscores the young leaves’ potential as a rich source of bioactive compounds with notable antioxidant and lipid-lowering properties, also providing valuable insights on the distribution of these compounds in different leaf stages.

### 3.3. In Vivo Experimental Study

#### 3.3.1. Effect of Treatments with Salicornia on the Animal Body and Organ Weights

The experiment progressed as scheduled, and no significant changes in the body weights of the animals were noted across the various treatment groups at each of the three time points of measurement. However, in the final measurement, the two groups subjected to CCl_4_ exhibited diminished values in comparison to the CTR and SAL groups, presumably attributed to the influence of the administered treatment (Figure 1). Additionally, there were no discernible differences in the mean liver and brain weights among the groups.

#### 3.3.2. The Ameliorative Effect of Salicornia on Plasma Biochemical Parameters

The investigation delved into the differences in hepatotoxicity, lipid profiles, and cardiovascular risk markers across the four experimental groups by conducting an in-depth plasma analysis, and the results were summarized in Table 3. All metabolic markers analyzed remained unaffected by administration of Salicornia alone, suggesting the non-toxicity of the prescribed dose over the fourteen days treatment period in SAL animals. Compared to controls, the CCl_4_ group exhibited significantly increased levels of AST and ALT, with the former being greater than the latter as typically occur following CCl_4_ exposure [8], indicating potential liver damage. CCl_4_-induced liver damage typically occurs through the process of its bioactivation into free radicals, specifically ^•^CCl_3_ and CCl_3_OO^•^ (trichloromethyl and trichloromethyl peroxyl free radicals). These radicals possess the capability to instigate various harmful intracellular and extracellular events. Reactive metabolites of CCl_4_ actively target and degrade polyunsaturated fatty acids, particularly those associated with phospholipids, leading to lipid peroxidation in cellular and organelle membranes that results in marked disruption of calcium homeostasis. As a result, necrotic cell death occurs due to the high permeability of plasma membranes to calcium ions and hepatocytes experience a pronounced disruption of cellular integrity, resulting in the substantial release of transaminases into the bloodstream. This event leads to a significant increase in ALT and AST levels, characteristic of hepatonecrosis [59], as observed in the present study. CCl_4_ + SAL group displayed significantly lower enzyme levels compared to CCl_4_ alone, suggesting a possible mitigating effect of the pre-treatment with Salicornia in the presence of carbon tetrachloride. A recent study by Taghipour et al. [60] showed how incorporating *Salicornia bigelovii* into the maintenance diet of another mammal species, the Shall sheep, could naturally boost antioxidant levels in the bloodstream, ultimately improving the animals’ overall health. Phenolic compounds naturally present in Halophytes derive their antioxidant efficacy through various mechanisms, such as scavenging free radicals [61] and quenching singlet-oxygen molecules [62]. Therefore, incorporating Salicornia into the diet may represent a natural means of increasing blood antioxidant levels, thus helping to improve animal health. In terms of lipid profile and cardiovascular risk markers, the CCl_4_ group showed a significant decrease in the levels of triglycerides, cholesterol, and the HDL-C and AAI index, together with an increase in TC/HDL-C, LDL-C/HDL-C and, although not significantly, LDL-C levels (Table 3), compared to controls. In response to the toxic effect of CCl_4_, the liver actively retrieves triglycerides from the bloodstream, resulting in the formation of lipid droplets, which leads to swelling in the liver tissues [63,64]. Previous research has confirmed the toxic effect of CCl_4_ and such liver reaction to counterbalance these metabolomic disorders [15,16,17]. However, the CCl_4_+SAL group’s values were all closer to those of the CTR and SAL groups, suggesting a potential protective influence of Salicornia. The perturbation of these blood parameters is widely recognized as a significant risk factor in the onset of various diseases, particularly cardiovascular ones. Similar to other natural compounds within the Halophyte plants, *S. europaea* has demonstrated efficacy in substantially improving deviations in blood parameters, thus preventing or reversing the progression of cardiovascular disease and TAFLD. Regarding renal function, the CCl_4_ group exhibited higher urea and creatinine levels, indicative of potential kidney dysfunction, while the SAL and CCl_4_+SAL groups demonstrated values more in line with the control group, suggesting an effect stabilizer of *S. europaea* on kidney function. Other halophyte extract showed similar protective effects in some previous studies [15,17]. The advantageous properties here observed can be mainly ascribed to the high polyphenolic content present in *S. europaea* extract, especially chlorogenic acid and 3-O-quercetin derivatives (Table 2). These compounds have previously demonstrated in vivo antioxidant efficacy and the capacity to reduce cholesterol levels through the modulation of PPAR-α gene expression, as evidenced by Wan et al. [65] and Grzelak-Błaszczyk et al. [66]. In summary, the data suggests the adverse effects of CCl_4_, with Salicornia clearly improving this detrimental impact.

#### 3.3.3. The Protective Effects of Salicornia on the Oxidative Stress Status

A comprehensive view of oxidative stress markers in the four different treated groups is shown in Figure 2. The CCl_4_-treated group exhibited a statistically significant increase in the levels of carbonylated proteins and thiobarbituric acid reactive substances (TBARS) and a decrease in GSH, compared to controls, indicating oxidative damage in the liver. As is well known, oxidative stress arises from an imbalance between the antioxidant and oxidant systems, with a proclivity towards the latter; disturbance of the normal redox state leads to the generation of free radicals, which can detrimentally impact biomolecules. This process affects cell membrane integrity through lipid peroxidation. In particular, MDA serves as a significant end product resulting from the free radical assault on polyunsaturated fatty acids within biological membranes, commonly employed to monitor lipid peroxidation [67]. Excessive accumulation of MDA, as observed in CCl_4_-treated rats, indicates the inability of endogenous antioxidant systems to halt the production of further toxic radicals. This, in turn, leads to progressive peroxidation and subsequent damage to hepatic tissues. In particular, some products of lipid peroxidation and the free radical generated can also affect tissue macromolecules, including proteins [15,17,53]. Compared to CCl_4_ group, the CCl_4_ + SAL group showed significantly lower values of carbonylated proteins and TBARS, comparable to those observed for CTR group, thus demonstrating that pretreatment with *S. europaea* was able to mitigate the oxidative stress induced by carbon tetrachloride [15,17]. These markers were not affected by the *S. europaea* treatment alone confirming that the dose administered for 14 days was not toxic for animals. Our results are in line with those recently presented by Ben Hsouna et al. [68], showing similar antioxidant effects from another halophyte plant, *Lobularia maritima*, against CCl_4_-induced hepatic oxidative damage in rat. In particular, in rats pre-treated with *L. maritima*, the authors observed a significant decrease in liver TBARS level and a decrease in hepatic marker enzyme levels in serum to their normal values. The advantageous attributes of antioxidants stem from their capacity to neutralize free radicals, commonly originating from molecules incorporating oxygen, nitrogen, or sulfur, collectively referred to as reactive free radicals (RRS) [69]. The EPR ‘radical probe’ technique was previously applied to assess the RRS yield in different animal and human tissues, as well as cell cultures [70,71], due to its high sensibility in measuring reactive radical species through all cell compartments. As depicted in Figure 3, no significant differences in RRS were observed between the control group and the Salicornia-treated group. In contrast, CCl_4_ administration led to an anticipated increase in RRS levels, resulting in a notable increase in the generation of radical species in both the liver and brain (Figure 3). CCl_4_ is able to easily traverse cell membranes, including the blood–brain barrier. Although all body tissues quickly absorb CCl_4_, its toxicity to the brain is not well comprehended. Intriguingly, the pretreatment with Salicornia demonstrated a significant reduction in free radicals’ levels, restoring them to the basal levels observed in the control and Salicornia groups. These findings are particularly noteworthy, affirming the robust antioxidant capacity of Salicornia against both hepatotoxicity and neurotoxicity. Notably, the almost neutral effect of Salicornia when it was administered to non-pathological animal units is of particular interest, considering that cellular redox homeostasis should be maintained. If on the one hand free radical scavenging is an effective means of cell protection from ROS-induced malignant transformation, conversely, too low ROS levels appear to contribute to the tumor progression and it can enhance the resistance of these cells to anticancer therapy [72]. A recent study by Zargar and Wani [73] demonstrated that quercetin, the second most represented phenolic compound in our extract (Table 2), was able to protect against CCl_4_-induced neurotoxicity in rats at given concentrations.

#### 3.3.4. The Effect of Salicornia on the Xenobiotic Metabolizing System

For the first time, we explored the potential preventive effect of *S. europaea* against the impact of CCl_4_ on the xenobiotic metabolizing system, examining the P450-dependent monooxygenase ECOD and pNPH, which serve, respectively, as indicators of multiple P450 isoforms, and CYP2E1 [7,41]. As illustrated in Figure 4, the administration of Salicornia at a dosage of 300 mg/kg bw did not produce any significant alteration in these hepatic enzyme’s activities, when compared to the control group. However, the CCl_4_ treatment resulted in a marked reduction in ECOD and pNPH linked-monooxygenases in comparison to both the control and Salicornia treated groups. It is conceivable that the recorded oxidative stress induced by intensive CCl_4_ metabolism by the P-450 superfamily of isoforms, and in particular by CYP2E1, determined a deleterious impact on cytochrome P450s. Previous research has indicated a decrease in numerous P450 specific activities following CCl_4_ administration, due to the well-established knowledge that ROS have a destructive effect on CYPs, leading to heme release. Concurrently, antioxidant enzymes, specifically Heme Oxygenase and DT-diaphorase, which are responsive to ROS, are likely to have been upregulated as a result [74]. Notably, in CCl_4_-intoxicated rats the pre-treatment with Salicornia determined a significant increase in both ECOD and pNPH, with the latter up to values comparable to those of the controls. In line with the findings related to P450-dependent oxidative metabolism, heme oxygenase and DT-diaphorase remained largely unaffected by Salicornia treatment alone, when compared to the control group (Figure 5). Conversely, CCl_4_ treatment led to a significant increase in heme oxygenase (Figure 5A), while DT-diaphorase showed only a slight and non-significant increase (Figure 5B). These results are consistent with previous studies, indicating the substantial impact of CCl_4_ on these enzyme activities [17,41]. Moreover, it is well-documented that CCl_4_ induces DT-diaphorase and heme oxygenase as an adaptive response to oxidative injury [15,16,17]. Notably, pre-treatment of rats with Salicornia appeared to partially mitigate the aforementioned changes. Overall, these findings suggest that Salicornia administration offers protection against liver damage. Diseases prevention through natural products is a topic of ongoing scientific discussion, and the administered dose of the substance can be a critical factor. In recent years, the scientific community has witnessed cases in which some compounds, initially proposed for disease prevention or risk reduction, have paradoxically increased the incidence of the very condition they were supposed to prevent [75,76].

#### 3.3.5. Hypolipidemic Effect of Salicornia

To assess the efficacy of Salicornia treatment in mitigating hepatic steatosis, we conducted measurements of total hepatic lipid content in separate study groups. As anticipated, rats treated with CCl_4_ showed significantly elevated hepatic lipid levels compared to both CTR and SAL groups, confirming the presence of steatosis. Remarkably, pretreatment with Salicornia (CCl_4_ + SAL) demonstrated a significant amelioration of the steatotic condition, evidenced by a significant reduction in hepatic lipid content, similar to the levels observed in the CTR and SAL groups (Figure 6A). To validate the structural impact of different treatments on liver tissue, we conducted a histological analysis using hematoxylin and eosin staining. As depicted in Figure 6B, liver sections from control animals showed healthy architecture with distinct nuclei and well-preserved cytoplasm, free of necrosis, inflammation, or steatosis evidence (Figure 6B, panels a and b). This normal architecture persisted in the liver tissue from the Salicornia-treated group, indicating that Salicornia alone did not adversely affect liver anatomy or function (Figure 6B, panels c and d). In contrast, CCl_4_-treated rats exhibited profound changes in liver morphology, including extensive macro and micro-vesicular steatosis and infiltration of inflammatory cells, probably of lymphocytic nature, concentrated around the central veins and portal system (arrows) (Figure 6B, panels e and f). This histopathological picture is consistent with a pattern of TAFLD [6]. For a long time, the buildup of fats in liver cells has been considered as a harmless toxicological finding. However, the ongoing epidemic of steatohepatitis and the growing connection with hepatocellular carcinoma have unavoidably prompted a revaluation of the importance of fatty liver diseases caused by environmental and industrial toxins. Fatty liver is now recognized as the most widespread pathological liver response to exposure to toxic substances. Uncertainties remain regarding the potential causal relationship between exposure to a toxicant and the onset of fatty liver disease, even though proposed potential mechanisms include mitochondrial dysfunction, impaired fatty acid exportation, elevated cytokine production, and insulin resistance. Moreover, it appears that nuclear receptors play a crucial role in the development of TAFLD. Activation of nuclear receptors occurs in response to exposure to foreign substances, leading to the induction of phase I and phase II drug-metabolizing enzymes. Specifically, it has been shown that peroxisome proliferator-activated receptors (PPARs), pregnane X receptors (PXR), constitutive androstane receptor (CAR), liver X receptor (LXR), farnesoid X receptor (FXR), and aryl hydrocarbon receptor (AHR) contribute to the onset of non-alcoholic fatty liver disease [77]. Noteworthy is that pretreatment with Salicornia alleviated the severity of both hepatic steatosis and inflammatory cell infiltration (Figure 6B, panels g and h). Such observations serve to corroborate the findings previously reported by other authors [15,16,17]. These distinct histopathological features observed among the four experimental groups align with the previously mentioned values of blood biochemical parameters, such as blood transaminases and blood fats (Table 3). Numerous studies have highlighted the antioxidant potential of plant extracts and polyphenol-rich vegetables in improving the hepatic lipid accumulation triggered by both fatty diets and toxicants [17,53]. Specifically, some isolated polyphenols, including phenolic acids and flavonoids such as chlorogenic acid and quercetin derivatives, have demonstrated efficacy in reducing liver lipid content in the presence of various toxicants and high-fat diets [15,17,53]. These beneficial compounds, abundantly found in halophyte plants, help improve fat metabolism in the liver [15,17]. Recent research explored the use of polyphenol-enriched extracts from the fruits and seeds *Sarcopoterium spinosum* in an in vitro model of hepatic steatosis, showcasing their potential to counteract lipid accumulation and improve hepato-steatosis [78]. Furthermore, much research has focused on the importance of extracts enriched with polyphenols and their implication in the molecular mechanisms of lipid-lowering action, as could be the case of Salicornia extract [53,78,79].

## 4. Conclusions

This study establishes the abundance of soluble polyphenols in Salicornia leaves extract affirming its status as a rich source of bioactive phenolic compounds, and highlighting, for the first-time, differential growth stage antioxidant capacity. Its ability to counteract the adverse effects associated with CCl_4_ treatment was demonstrated by the clear efficacy of Salicornia pretreatment in an in vivo model of TAFLD. As evidenced in our investigation, the inclusion of polyphenol-rich foods, such as Salicornia, in dietary habits is related to a spectrum of health benefits, which include the improvement of serum biochemical parameters, antioxidant properties, and the ability to mitigate liver steatosis by reducing lipid levels in animals. In light of these findings, Salicornia emerges as a valuable component in meal planning, especially for individuals struggling with cardiovascular disease, liver disorders, and steatosis. However, recognizing the complexity of these interactions, further research is essential to unravel the intricate mechanisms that underlie the observed health improvements. Specifically, a deeper understanding of the complex relationship between polyphenols and molecular pathways governing TAFLD becomes crucial, given the positive impact of Salicornia’s polyphenols on oxidative status and lipid lowering observed in our study. Moreover, limitations associated with the current study include the absence of assessment of the observed effects on females to determine any potential sex-specific responses, as well as the lack of measurement of the content or effects on fat-soluble antioxidants such as Vitamin E.

## Figures and Tables

**Figure 1 antioxidants-13-00596-f001:**
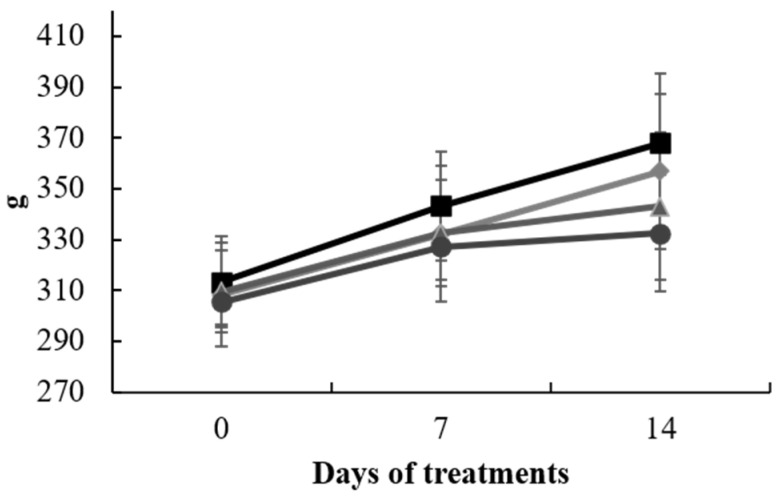
Animal body weight (g) from 0 to 14 days of treatment with *S. europaea*. Data represent the mean ± SD of each animal from control group (□), SAL (◊), CCl_4_ (∆), and CCl_4_ + SAL (○) treated groups (n = 10).

**Figure 2 antioxidants-13-00596-f002:**
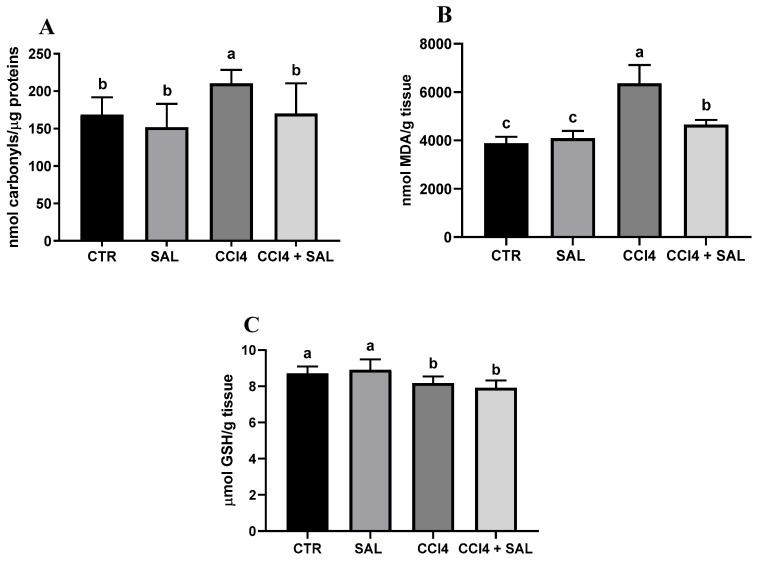
Protein carbonylation (**A**), malondialdehyde (MDA) (**B**), and GSH content (**C**) in the liver of all CTR, SAL, CCl_4_, and CCl_4_ + SAL rats, performing three replicates for each animal. Values are expressed as means ± SD (n = 10). Values within each row different letters (a,b,c) are significantly different with a one way ANOVA-test (*p* ≤ 0.05).

**Figure 3 antioxidants-13-00596-f003:**
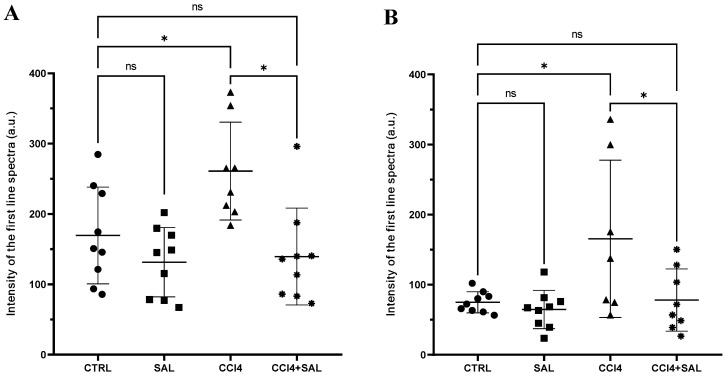
Free radical measurements through EPR spectroscopy in the liver (**A**) and brain (**B**) of all CTR (•), SAL (■), CCl_4_ (▲), and CCl_4_ + SAL (*) rats. The scattered dot plot represents total oxygen-, nitrogen-, and carbon-centered free radical species (arbitrary units). Values are expressed as means ± SD (panel **A**: CTR, SAL, CCl_4_ + SAL n = 9, CCl_4_ n = 8; panel **B**: CTR, SAL n = 9, CCl_4_ n = 7, CCl_4_ + SAL n = 8). *: values significantly different with a one-way ANOVA-test (*p* ≤ 0.05); ns: not significant.

**Figure 4 antioxidants-13-00596-f004:**
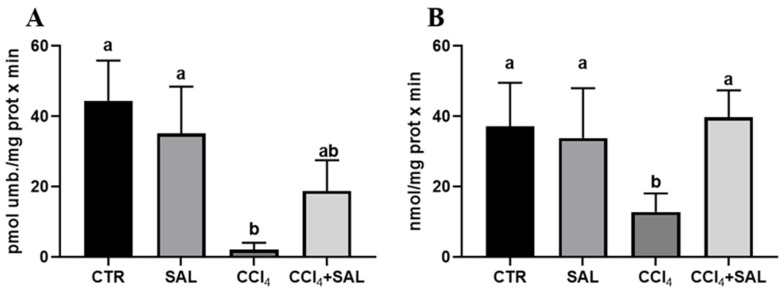
Ethoxycoumarin-O-deethylase, ECOD (**A**) and p-nitrophenol hydroxylase, pNPH (**B**) activities in liver of all CTR, SAL, CCl_4_, and CCl_4_ + SAL rats, performing three replicates for each animal. Values are expressed as means ± SD (n = 10) and reported as pmol umbelliferone/(mg protein × min) for ECOD and as nmol/(mg protein × min) for pNPH. a, b: values significantly different with a one-way ANOVA-test (*p* ≤ 0.05).

**Figure 5 antioxidants-13-00596-f005:**
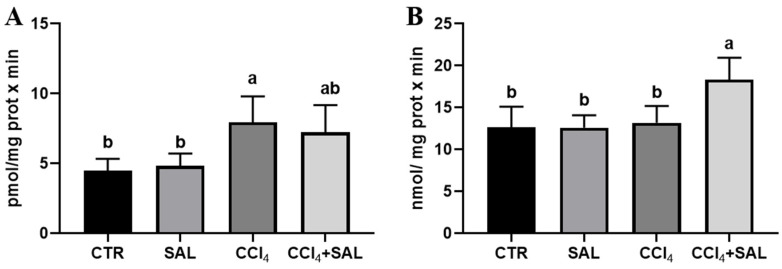
Heme oxygenase (**A**) and DT-diaphorase (**B**) activities in liver of all CTR, SAL, CCl_4_, and CCl_4_ + SAL rats, performing three replicates for each animal. Values are expressed as mean ± SD of rats in each group (n = 10) and reported as pmol/(mg protein × min) for heme oxygenase and as nmol/(mg protein × min) for DT-diaphorase. a, b: values significantly different with a one-way ANOVA-test (*p* ≤ 0.05).

**Figure 6 antioxidants-13-00596-f006:**
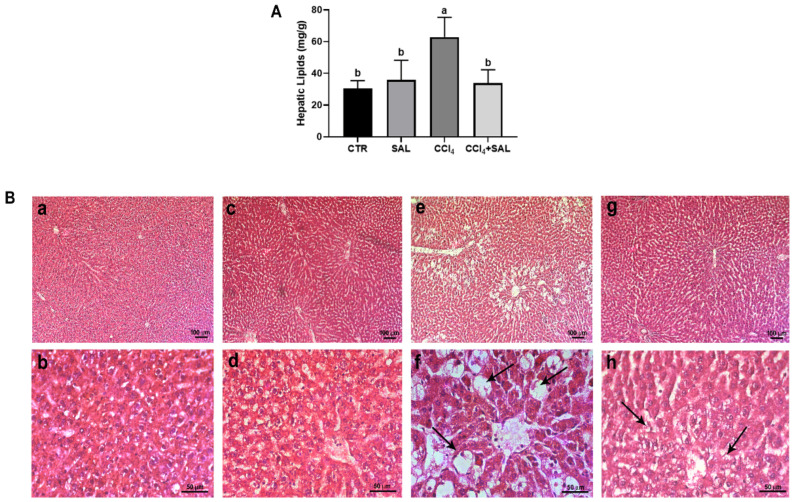
Total hepatic lipid content (**A**) measured in liver tissue from all CTR, SAL, CCl_4_, and CCl_4_ + SAL rats, performing three replicates for each animal. Values are expressed as means ± SD (n = 10). a, b: values significantly different with a one-way ANOVA-test (*p* ≤ 0.05). Hematoxylin and eosin (H&O) staining (**B**) of liver tissue from CTR (**a**,**b**), SAL (**c**,**d**), CCl_4_ (**e**,**f**), and CCl_4_ + SAL (**g**,**h**) rats. Arrows indicate lipids ballooning (**f**) or a decrease in steatosis (**h**). Magnification: (**a**,**c**,**e**,**g**), bar = 100 µm; (**b**,**d**,**f**,**h**), bar = 50 µm.

**Table 1 antioxidants-13-00596-t001:** Bioactive compounds and in vitro antioxidant activity of young and old leaf extracts of *S. europaea*.

		*S. europaea* Leaves Extract	
		Young Leaves	Old Leaves
Bioactive compounds	Total polyphenols (mg GAE/g DW)	8.53 ^a^ ± 0.41	4.26 ^b^ ± 0.12
	Total flavonoids (mg CE/g DW)	0.51 ^a^ ± 0.08	0.14 ^b^ ± 0.05
	Flavonols (mg QE/g DW)	0.11 ^a^ ± 0.01	0.05 ^b^ ± 0.01
	Anthocyanins (mg C3GE/g DW)	0.18 ^a^ ± 0.01	0.10 ^b^ ± 0.00
Pigments	Chlorophyll a (µg Chl a/g FW)	465.30 ^a^ ± 10.43	407.80 ^b^ ± 9.97
	Chlorophyll b (µg Chl b/g FW)	233.40 ^a^ ± 15.48	157.80 ^b^ ± 12.14
	Carotenoids (µg/g FW)	106.70 ^a^ ± 2.12	102.6 ^a^ ± 1.73
Antioxidant activity	DPPH (ARA %)	79.51 ^a^ ± 0.45	60.57 ^b^ ± 0.33
	FRAP (EC50, mg/mL)	6.79 ^a^ ± 0.44	3.37 ^b^ ± 0.10
	ORAC (µmol TE/g DW)	268.08 ^a^ ± 21.10	164.88 ^b^ ± 29.13

Values are reported as means ± SD of three replicates (n = 3). Values within each row different letters (a,b) are significantly different with Student’s *t*-test (*p* ≤ 0.05).

**Table 2 antioxidants-13-00596-t002:** Content of individual phenolic compounds in the hydro-methanolic extract of young and old leaves of *S. europaea*.

Compound Name	**Content in** ** *Salicornia europeae* ** **(µg/g DW)**	
	Young Leaves	Old Leaves
Gallic acid	0.69 ^a^ ± 0.01	0.11 ^b^ ± 0.00
3-*O*-Caffeoylquinic acid (chlorogenic acid)	1984.85 ^a^ ± 0.11	865.20 ^b^ ± 0.16
Rosmarinic acid	0.22 ^a^ ± 0.00	0.16 ^b^ ± 0.00
Caffeic acid	3.18 ^a^ ± 0.04	0.52 ^b^ ± 0.01
*p*-Coumaric acid	1.01 ^a^ ± 0.01	0.38 ^b^ ± 0.00
Vanillic acid	16.55 ^a^ ± 0.14	5.14 ^b^ ± 0.05
*trans*-Ferulic acid	39.13 ^a^ ± 0.07	7.52 ^b^ ± 0.07
Protocatechuic acid	4.52 ^a^ ± 0.04	0.36 ^b^ ± 0.00
**∑ Phenolic acids ***	**2050.15**	**879.39**
(+)-Catechin	0.53 ^a^ ± 0.00	0.30 ^b^ ± 0.00
(−)-Epicatechin	0.61 ^a^ ± 0.00	0.30 ^b^ ± 0.00
Apigenin	0.04 ^b^ ± 0.00	0.07 ^a^ ± 0.00
**∑** **Flavan-3-ols ***	**1.18**	**0.67**
Naringenin	2.49 ^a^ ± 0.02	0.04 ^b^ ± 0.00
Eriodictyol	0.02 ^b^ ± 0.00	0.11 ^a^ ± 0.00
**∑ Flavanones ***	**2.51**	**0.15**
Quercetin	0.16 ^b^ ± 0.00	0.27 ^a^ ± 0.00
Kaempferol 3-*O*-glucoside	0.05 ^b^ ± 0.00	0.08 ^a^ ± 0.00
Kaempferol 3-*O*-rutinoside	12.87 ^a^ ± 0.05	3.88 ^b^ ± 0.03
Quercetin 3-*O*-glucoside	154.33 ^a^ ± 0.05	61.99 ^b^ ± 0.08
Quercetin 3-*O*-rutinoside (rutin)	7.80 ^a^ ± 0.04	6.87 ^b^ ± 0.09
Quercetin 3,4-*O*-diglucoside	0.32 ^a^ ± 0.00	Nd
**∑ Flavonols ***	**175.53**	**73.09**
Resveratrol	0.16 ^a^ ± 0.00	0.12 ^b^ ± 0.00
Resveratrol 3-*O*-glucoside (piceid)	Nd	0.04 ^a^ ± 0.00
**∑** **Stilbenoids ***	**0.16**	**0.16**
Hydroxytyrosol	0.41 ^a^ ± 0.00	0.13 ^b^ ± 0.00
Luteolin	0.21 ^b^ ± 0.00	0.29 ^a^ ± 0.00
Oleuropein	0.06 ^a^ ± 0.00	0.03 ^b^ ± 0.00
Verbascoside	0.03 ^b^ ± 0.00	0.04 ^a^ ± 0.00
Phloretin	0.03 ^a^ ± 0.00	0.02 ^b^ ± 0.00
Phloridzin	0.01 ^a^ ± 0.00	0.01 ^a^ ± 0.00
**∑** **Others ***	**0.75**	**0.52**

Values are reported as means ± SD of three replicates (n = 3). Values within each row different letters (a,b) are significantly different with a student *t*-test (*p* ≤ 0.05). Nd: not detected. * Sum of phenolic acids, flavan-3-ols, flavanones, flavonols, stilbenoids, and other compounds determined using HPLC-MS.

**Table 3 antioxidants-13-00596-t003:** Biochemical parameters in the rat plasma of CTR, SAL, CCl_4_, and CCl_4_ + SAL.

	CTR (n = 10)	SAL (n = 9)	CCl_4_ (n = 10)	CCl_4_ + SAL (n = 9)
AST (U/L)	121.9 ^b^ ± 42.3	110.3 ^b^ ± 35.6	1894.2 ^a^ ± 217.0	181.7 ^b^ ± 93.0
ALT (U/L)	43.6 ^b^ ± 2.9	42.5 ^b^ ± 10.0	1172.2 ^a^ ± 123.0	54.2 ^b^ ± 8.0
TC (mg/dL)	85.2 ^a^ ± 12.8	80.8 ^a^ ± 7	37.2 ^b^ ± 7.9	70.3 ^a^ ± 14.1
HDL-C (mg/dL)	37.4 ^a^ ± 4.1	38.7 ^a^ ± 4.7	12.3 ^b^ ± 7.1	29.3 ^a^ ± 5.0
LDL-C (mg/dL)	6.8 ^a^ ± 2.2	6.4 ^a^ ± 1.1	7.2 ^a^ ± 2.7	6.2 ^a^ ± 1.6
TG (mg/dL)	166.7 ^a^ ± 47.6	139.1 ^a^ ± 33.6	26.3 ^c^ ± 10.2	74.3 ^b^ ± 39.6
Urea (mg/dL)	37.4 ^b^ ± 4.6	37.8 ^b^ ± 5.2	55.3 ^a^ ± 6.1	51.5 ^a^ ± 5.7
Creatinine (mg/dL)	0.28 ^b^ ± 0.03	0.27 ^b^ ± 0.05	0.39 ^a^ ± 0.01	0.30 ^b^ ± 0.03
TC/HDL-C (ratio)	2.3 ^b^ ± 0.4	2.1 ^b^ ± 0.1	3.8 ^a^ ± 1.8	2.4 ^b^ ± 0.3
LDL-C/HDL-C (ratio)	0.18 ^b^ ± 0.07	0.17 ^b^ ± 0.04	0.92 ^a^ ± 0.82	0.21 ^b^ ± 0.05
AAI *	81.9 ^a^ ± 18.0	92.6 ^a^ ± 9.7	50.8 ^b^ ± 28.9	73.6 ^ab^ ± 13.1

Results are reported as means ± SD of three replicates (n = 3). Values within each row different letters (a,b,c) are significantly different with a one-way ANOVA (*p* ≤ 0.05). * Antiatherogenic index (AAI) = HDL-C × 100/TC − HDL-C.

## Data Availability

The original data will be accessible to individuals upon request to the corresponding author.

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
