# Peer review of "Assessment of the Antioxidant and Hypolipidemic Properties of *Salicornia europaea* for the Prevention of TAFLD in Rats"

_antioxidants, 2024, doi:10.3390/antiox13050596_

Round 1
Reviewer 1 Report
I believe that this is a very interesting and well executed, in depth, analysis of constituents and the properties and the protective and antioxidant effects of the extract of Salicornia europaea.
I gladly propose the acceptance of the manuscript after some modifications.
Some issues that need to be addressed and to be taken into account are listed below.
1. Please analyze thoroughly the in vitro assays of antioxidant activity, according to the literature that is referred.
2. How was the route and the dose of administration of CCl4 selected? Please refer to the analogous literature.
3. Please make an in detail review of the manuscript for potential typographical or figure errors or omissions (e.g. in Figure 2 A its carbonyls)
4. The authors used maceration in 10 mL of 80% methanol. Why this solvent and this proportion of solvent chosen. Could and other solvents be used in order to compare the activities of the extracts? If these solvents were chosen based on a previous research, a relevant reference should be added.
5. Please refer to the full title of acronyms at the first time of appearance e.g. CFCs
6. In some places the parenthesis are missing e.g. (alanine transaminase (ALT)
I believe that this is a very interesting and well executed, in depth, analysis of constituents and the properties and the protective and antioxidant effects of the extract of Salicornia europaea.
I gladly propose the acceptance of the manuscript after some modifications.
Some issues that need to be addressed and to be taken into account are listed below.
1. Please analyze thoroughly the in vitro assays of antioxidant activity, according to the literature that is referred.
2. How was the route and the dose of administration of CCl4 selected? Please refer to the analogous literature.
3. Please make an in detail review of the manuscript for potential typographical or figure errors or omissions (e.g. in Figure 2 A its carbonyls)
4. The authors used maceration in 10 mL of 80% methanol. Why this solvent and this proportion of solvent chosen. Could and other solvents be used in order to compare the activities of the extracts? If these solvents were chosen based on a previous research, a relevant reference should be added.
5. Please refer to the full title of acronyms at the first time of appearance e.g. CFCs
6. In some places the parenthesis are missing e.g. (alanine transaminase (ALT)
Author Response
- Please analyze thoroughly the in vitro assays of antioxidant activity, according to the literature that is referred.
Done. Each in vitro test method has been described in paragraph 2.2.3 of the text.
- How was the route and the dose of administration of CCl4 selected? Please refer to the analogous literature.
The intraperitoneal (i.p.) route for CCl4 was decided for its effectiveness and safety of administration on the basis of some previous studies performed in our laboratories (Longo et al., 2007. https://doi.org/10.1007/s10529-007-9378-6; Souid at al., 2020)
The dose of 0.8 mL/kg bw was selected for being a relatively low dosage with the respect to the existing literature for in vivo rat studies, as for example our previous studies (see above). This dose would ensure the onset of steatosis while avoiding acute liver failure (Dai et al., 2014; De Minicis et al., 2014). Previous studies revealed to be strictly dependent on exposure time and dosage since CCl4 can induce fatty liver very rapidly. As explained in Introduction section, our choice arose from the necessity to explore the consequences of minimal exposure to hazardous compounds, as such exposure, even at low concentrations typical of daily life, holds the potential to trigger steatosis and liver damage, as it happens in TAFLD.
- Please make an in-detail review of the manuscript for potential typographical or figure errors or omissions (e.g. in Figure 2 A its carbonyls)
Done.
- The authors used maceration in 10 mL of 80% methanol. Why this solvent and this proportion of solvent chosen. Could and other solvents be used in order to compare the activities of the extracts? If these solvents were chosen based on previous research, a relevant reference should be added.
The selection of 80% methanol as the preferred solvent for extracting S. europaea was based on a prior investigation conducted by Qasim et al. in 2016 on Halophytes plants. This study showed that this specific solvent concentration enables optimal extraction of polyphenols and exhibits superior antioxidant activity compared to all alternative solvents such as ethanol, water, acetone, and chloroform, and also compared to different percentages of the same solvent [Qasim, M., Aziz, I., Rasheed, M., Gul, B., & Khan, M. A. (2016). Effect of extraction solvents on polyphenols and antioxidant activity of medicinal halophytes. Pak. J. Bot, 48(2), 621-627]. The reference has been added to the text.
- Please refer to the full title of acronyms at the first time of appearance e.g. CFCs
Done.
- In some places the parenthesis are missing e.g. (alanine transaminase (ALT)
The errors have been corrected in the text.
All the revisions suggested by the reviewer 1 have been highlighted in the text in yellow.

Reviewer 2 Report
The studies summarized in the manuscript demonstrate good antioxidant properties of extracts of leaves of
Salicornia europaea, their hypolipidemic properties and protection against carbon-tetrachloride-induced liver injury in experimental animals administered previously with the extract. Increased salinity imposes oxidative stress so halophytes are expected to be rich in antioxidants. Of course, any living tissue and, especially, any plant material has antioxidant properties. This was demonstrated and analyzed in this study for S. europaea. I wonder whether it is possible to demonstrate that S. europaea is richer in antioxidants than non-halophyte species, on the basis of literature or own data? The authors cite data concerning other halophytes but do not compare them with non-halophyte plants.
The analysis of antioxidant activity is supported by analysis of polyphenol, flavonoid, flavonols, anthocyanin and carotenoid content, and results of HPLC analysis of phenolic compounds.
The demonstration of the higher antioxidant activity and higher content of antioxidant compounds in young and mature leaves is interesting. The results of animal experiments convincingly demonstrate the beneficial effects of S. europaea extracts.
The Lines 114, 116, 120, 123, 125: S. europaea, please in italics,
Lines129-133: This question seems obvious but were there any objective, measurable criteria to characterize leaves as mature?
Line 215: “at 4,000 rpm”, please report rather the g value, rpm effects are different depending on the centrifuge geometry. Otherwise, please specify the centrifuge type and rotor.
Line 251:” TEMPO-coline” or TEMPO-choline?
Line 434:” heigh permeability’, high permeability?
Line 445: : quenching singlet-oxygen atoms”, singlet oxygen is a molecule, not an atom
Line 458:” within the Halophyte family”, halophytes are not family in the taxonomic sense
Please explain ARA
Author Response
The studies summarized in the manuscript demonstrate good antioxidant properties of extracts of leaves of Salicornia europaea, their hypolipidemic properties and protection against carbon-tetrachloride-induced liver injury in experimental animals administered previously with the extract. Increased salinity imposes oxidative stress, so halophytes are expected to be rich in antioxidants. Of course, any living tissue and, especially, any plant material has antioxidant properties. This was demonstrated and analyzed in this study for S. europaea. I wonder whether it is possible to demonstrate that S. europaea is richer in antioxidants than non-halophyte species, on the basis of literature or own data? The authors cite data concerning other halophytes but do not compare them with non-halophyte plants.
Interestingly, the value of total phenols found for S. europaea in the present study are comparable to those previously found in our laboratory for a non-halophyte species, in particular a variety of Brassica oleracea leaves (Kavolì) (9.23 ± 0.32 mg GAE/g dw) [Vornoli et al., 2022]. Brassicas are notoriously vegetables with a very high antioxidant content, and this further supports the results found for S.europaea.
The analysis of antioxidant activity is supported by analysis of polyphenol, flavonoid, flavonols, anthocyanin and carotenoid content, and results of HPLC analysis of phenolic compounds.
The demonstration of the higher antioxidant activity and higher content of antioxidant compounds in young and mature leaves is interesting. The results of animal experiments convincingly demonstrate the beneficial effects of S. europaea extracts.
Detail comments
The Lines 114, 116, 120, 123, 125: S. europaea, please in italics
Done.
Lines129-133: This question seems obvious but were there any objective, measurable criteria to characterize leaves as mature?
As described in lines 133-136, the following are the visual characteristics and growth stage that permit to easily distinguish young from adult leaves: young Salicornia shoots (leaves) are tender, slender, and green, boasting an elegant appearance that makes them a sought-after choice for culinary use. In contrast, adult shoots are more mature, thicker, and may develop a woody texture, often acquiring reddish or brownish hues with age.
Furthermore, objective, measurable criteria have been characterized for the first time in the present work regarding detailed phenolic compounds profiling and antioxidant potential.
Line 215: “at 4,000 rpm”, please report rather the g value, rpm effects are different depending on the centrifuge geometry. Otherwise, please specify the centrifuge type and rotor.
Relative centrifugal force (RCF), or g force, value has been substituted to RPM value and corrected in the text on the basis of centrifuge geometry.
Line 251:” TEMPO-coline” or TEMPO-choline?
TEMPO-coline.
Line 434:” heigh permeability’, high permeability?
The typing error has been corrected.
Line 445: quenching singlet-oxygen atoms”, singlet oxygen is a molecule, not an atom
The error has been corrected.
Line 458:” within the Halophyte family”, halophytes are not family in the taxonomic sense
“family” has been substituted with “plants”.
Please explain ARA
Done. “ARA” (antiradical activity) was explained in the text (paragraph 2.2.3)
All the revisions suggested by the reviewer 2 have been highlighted in the text in green.

Reviewer 3 Report
This study highlights Salicornia's potential as a source of bioactive compounds with effective hepatoprotective properties that are capable of preventing low-dose CCl4-induced toxicant-associated fatty liver disease (TAFLD). The article is interesting and novel because it characterizes various phenolic compounds in young leaves compared to mature ones. Salicornia treatment mitigated 2CCl4-induced toxicant-associated fatty liver disease (TAFLD) oxidative stress, reduced lipid peroxidation, and protein carbonyl levels, and preserved the decrease in glutathione levels.
No significant concerns were spotted during the review process; it would be helpful if the authors compared the methanol-extracted polyphenols to those of other methods of antioxidant extraction. The tables and figures are well organized, and the most bioactive compounds measured in the study show that the effect of aging of plants is to diminish the levels of the antioxidants contained in the leaves. It might be helpful to include the information for the sample size (n=3, five or more) in each table or the figure legend. Figure 1 is not very valuable because it starts at 270g and goes to 410g, and there is no significant statistical effect of treatment with SAL (probably, weight changes will be better in table format). Some of the figures are organized as simple bar graphs (Fig 2, 4, 5, and 6a), and one figure (Fig 4) is in the format of a scattered dot plot. It would be helpful for the readers if all figures were in the same scatter dot plot format (flowing the way data is present in Fig 4). One additional area for improvement will be to add a list of limitations (for example, the study used only male animals, and there is no data for females; the authors did not measure the effects or content of fat-soluble antioxidants such as Vitamin E and others).
Author Response
This study highlights Salicornia's potential as a source of bioactive compounds with effective hepatoprotective properties that are capable of preventing low-dose CCl4-induced toxicant-associated fatty liver disease (TAFLD). The article is interesting and novel because it characterizes various phenolic compounds in young leaves compared to mature ones. Salicornia treatment mitigated 2CCl4-induced toxicant-associated fatty liver disease (TAFLD) oxidative stress, reduced lipid peroxidation, and protein carbonyl levels, and preserved the decrease in glutathione levels.
Detail comments
No significant concerns were spotted during the review process; it would be helpful if the authors compared the methanol-extracted polyphenols to those of other methods of antioxidant extraction.
The selection of 80% methanol as the preferred solvent for extracting S. europaea was based on a prior investigation conducted by Qasim et al. in 2016 on Halophytes plants. This study showed that this specific solvent concentration enables optimal extraction of polyphenols and exhibits superior antioxidant activity compared to all alternative solvents such as ethanol, water, acetone, and chloroform, and also compared to different percentages of the same solvent [Qasim, M., Aziz, I., Rasheed, M., Gul, B., & Khan, M. A. (2016). Effect of extraction solvents on polyphenols and antioxidant activity of medicinal halophytes. Pak. J. Bot, 48(2), 621-627]. The reference has been added to the text.
The tables and figures are well organized, and the most bioactive compounds measured in the study show that the effect of aging of plants is to diminish the levels of the antioxidants contained in the leaves. It might be helpful to include the information for the sample size (n=3, five or more) in each table or the figure legend.
Done. In each table/figure’s caption, the number of samples (n=x) and also replicates (usually three for each animal parameter) have been added.
Figure 1 is not very valuable because it starts at 270g and goes to 410g, and there is no significant statistical effect of treatment with SAL (probably, weight changes will be better in table format).
The depicted graph facilitates the reader's verification of the similarity of all groups in trends across different time points. However, in comparison to a table, it enables us to better discern that the two groups treated with CCl4 exhibited less growth between the second and third time points than the other two groups. Therefore, this graph helps to understand how treatment with CCl4, although not significantly, influences and slows down the growth of the animal, as described in paragraph 3.3.1.
Some of the figures are organized as simple bar graphs (Fig 2, 4, 5, and 6a), and one figure (Fig 4) is in the format of a scattered dot plot. It would be helpful for the readers if all figures were in the same scatter dot plot format (flowing the way data is present in Fig 4).
Free radical measurements through EPR spectroscopy, due to the complexity of the method, during experimental procedure resulted in the loss of some units within groups C and D, as specified in the caption of Figure 3. Therefore, we opted to use scattered dot plots to display the actual sample sizes (and their distribution) in each group. Thankfully, in all the other experiments, the sample sizes were always consistent (n=10), and we did not find it necessary to present the results as distributions in two-dimensional space. Instead, we represented them as bars with standard deviations as is customary for presenting this type of results in the scientific community.
One additional area for improvement will be to add a list of limitations (for example, the study used only male animals, and there is no data for females; the authors did not measure the effects or content of fat-soluble antioxidants such as Vitamin E and others).
In the Conclusions, an additional section highlighting the suggested limitations has been included.
All the revisions suggested by the reviewer 3 have been highlighted in the text in light blue.
